# LEARNING SPECIALIZED ACTIVATION FUNCTIONS FOR PHYSICS-INFORMED NEURAL NETWORKS

## ABSTRACT

At the heart of network architectures lie the non-linear activation functions, the choice of which affects the model optimization and task performance. In computer vision and natural language processing, the Rectified Linear Unit is widely adopted across different tasks. However, there is no such default choice of activation functions in the context of physics-informed neural networks (PINNs). It is observed that PINNs exhibit high sensitivity to activation functions due to the various characteristics of each physics system, which makes the choice of the suitable activation function for PINNs a critical issue. Existing works usually choose activation functions in an inefficient trial-and-error manner. To address this problem, we propose to search automatically for the optimal activation function when solving different PDEs. This is achieved by learning an adaptive activation function as linear combinations of a set of candidate functions, whose coefficients can be directly optimized by gradient descent. In addition to its efficient optimization, the proposed method enables the discovery of novel activation function and the incorporation with prior knowledge about the PDE system. We can further enhance its search space with adaptive slope. While being surprisingly simple, the effectiveness of the proposed adaptive activation function is demonstrated on a series of benchmarks, including the Poisson's equation, convection equation, Burgers' equation, Allen-Cahn equation, Korteweg–de Vries equation, and Cahn-Hilliard equation. The performance gain of the proposed method is further interpreted from the neural tangent kernel perspective. Code will be released.

## 1 INTRODUCTION

Recent years have witnessed the remarkable progress of physics-informed neural networks (PINNs) on simulating the dynamics of physical systems, in which the activation functions play a significant role in the expressiveness and optimization of models. While the Rectified Linear Unit (Hahnloser et al., 2000; Jarrett et al., 2009; Nair & Hinton, 2010) is widely adopted in most computer vision and natural language processing tasks (Ramachandran et al., 2017), there is no such default choice of activation functions in the context of PINNs. In fact, PINNs show great sensitivity to activation functions when applied to different physical systems, since each system has its own characteristic. On one hand, the utilization of unsuitable activation functions would cause over-parameterization and overfitting. One the other hand, accurate simulations can be achieved with fast convergence and high precision by choosing proper activation functions. For example, the hyperbolic tangent function is shown to suffer from numerical instability when simulating vortex induced vibrations, while a PINN with sinusoidal function can be optimized smoothly (Raissi et al., 2019b).

The various characteristics of different PDE systems make it critical to select proper activation functions in PINNs. The common practice to find the optimal activation functions is by trial-and-error, which requires extensive computational resources and human knowledge. This method is inefficient especially when solving complex problems, where searching for a set of activation functions is necessary to make accurate predictions. For instance, a combination of the sinusoidal and the exponential function is demonstrated effective to solve the heat transfer equation, whose solution is periodic in space and exponentially decaying in time (Zobeiry & Humfeld, 2021). In this case, the trial-and-error strategy leads to the combinatorial search problem, which becomes infeasible when the candidate activation function set is large.

In this work, we propose a simple and effective physics-informed activation function (PIAC), aiming at the automatic design of activation functions for solving PDE systems with various characteristics. Sharing the same spirits with differentiable neural architecture search (Liu et al., 2018), we parameterize the categorical selection of one particular activation function into a continuous search space, leading to an end-to-end differentiable problem which can be integrated into the training procedure of PINNs. Specifically, we first define a set of candidate activation functions, which is conceptually composed of simple elementary functions or commonly-used activation functions. Then, the proposed PIAC is learned as the linear combination of candidate functions with adaptive coefficients. Besides its efficient optimization, this continuous parameterization also enables the discovery of novel activation functions, whose capacity can be further enhanced by cascading these learnable functions in a layer-wise or neuron-wise manner.

Our method can be regraded as adaptive activation functions built on a predefined candidate function set. Although adaptive activation functions have been explored for PINNs, previous works mainly focus on accelerating the convergence of PINNs by introducing the adaptive slope (Jagtap et al., 2020b;a), while leaving the inefficient selection of activation function for different PDE systems unexplored. In fact, our method is orthogonal to these methods and can be extended by incorporating the adaptive slope into our search space. Moreover, the introduction of candidate function set can be leveraged to embed prior knowledge about the PDE system into the neural networks. For example, the sinusoidal function can be added into this set to assist the modeling of periodicity.

We evaluate the proposed PIAC on a series of PDEs, including the Poisson's equation, convection equation, Burgers' equation, Allen-Cahn equation, Korteweg–de Vries equation, and Cahn-Hilliard equation. Extensive experiments show that the proposed PIAC can consistently outperform the standard activation functions. We further explain the performance gain from the neural tangent kernel (NTK) perspective (Jacot et al., 2018; Wang et al., 2022).

The main contribution of this paper can be summarized as:

- We investigate the influence of activation functions for PINNs to solve different problems and reveal the high sensitivity of PINNs to the choice of activation functions, which can be related to various characteristics of the underlying PDE system;

- We explore the automatic design of activation functions for PINNs. The proposed method can be efficiently optimized and adapted to the PDE system, while enabling the utilization of prior knowledge about the problem;

- While being simple, the effectiveness of the proposed PIAC is demonstrated on extensive experiments and interpreted from the perspective of neural tangent kernel.

## 2 PRELIMINARIES AND RELATED WORK

### 2.1 PHYSICS-INFORMED NEURAL NETWORKS

Physics-informed neural networks have emerged as a promising method for solving forward and inverse problems of PDEs (Raissi et al., 2019a; Chen et al., 2020; Lu et al., 2021b; Karniadakis et al., 2021), fractional PDEs (Pang et al., 2019), and stochastic PDEs (Zhang et al., 2019). In this work, we focus on solving the forward problems of PDEs as described in (Raissi et al., 2019a). Specifically, we consider PDEs of the general form

$$\boldsymbol{u}_t + \mathcal{N}[\boldsymbol{u}(t, \boldsymbol{x}); \lambda] = 0, \ \boldsymbol{x} \in \Omega \subset \mathbb{R}^d, \ t \in [0, T], \tag{1}$$

subject to the initial and boundary conditions

$$\boldsymbol{u}(0, \boldsymbol{x}) = \boldsymbol{u}^0(\boldsymbol{x}), \ \boldsymbol{x} \in \Omega, \tag{2}$$

$$\mathcal{B}[\boldsymbol{u}] = 0, \ \boldsymbol{x} \in \partial\Omega, \ t \in [0, T], \tag{3}$$

where $\boldsymbol{u}(t, \boldsymbol{x})$ denotes the solution, $\mathcal{N}[\cdot; \lambda]$ is a differential operator parameterized by $\lambda$, $\mathcal{B}[\cdot]$ is a boundary operator, and subscripts denote the partial differentiation.

A physics-informed neural network (PINN) $\boldsymbol{u}'(t, \boldsymbol{x}; \boldsymbol{\theta})$ is optimized to approximate the solution $\boldsymbol{u}(t, \boldsymbol{x})$ by minimizing the following objective function

$$\mathcal{L}(\boldsymbol{\theta}) = \mathcal{L}_{\text{ic}}(\boldsymbol{\theta}) + \mathcal{L}_{\text{bc}}(\boldsymbol{\theta}) + \mathcal{L}_{\text{r}}(\boldsymbol{\theta}), \tag{4}$$

where

$$\mathcal{L}_{\text{ic}}(\boldsymbol{\theta}) = \frac{1}{N_{\text{ic}}} \sum_{i=1}^{N_{\text{ic}}} ||\boldsymbol{u}'(0, \boldsymbol{x}_{\text{ic}}^i; \boldsymbol{\theta}) - \boldsymbol{u}^0(\boldsymbol{x}_{\text{ic}}^i)||_2^2, \tag{5}$$

$$\mathcal{L}_{\text{bc}}(\boldsymbol{\theta}) = \frac{1}{N_{\text{bc}}} \sum_{i=1}^{N_{\text{bc}}} ||\mathcal{B}[\boldsymbol{u}'(t_{\text{bc}}^i, \boldsymbol{x}_{\text{bc}}^i; \boldsymbol{\theta})]||_2^2, \tag{6}$$

$$\mathcal{L}_{\text{r}}(\boldsymbol{\theta}) = \frac{1}{N_{\text{r}}} \sum_{i=1}^{N_{\text{r}}} ||\boldsymbol{u}_t'(t_{\text{r}}^i, \boldsymbol{x}_{\text{r}}^i; \boldsymbol{\theta}) + \mathcal{N}[\boldsymbol{u}'(t_{\text{r}}^i, \boldsymbol{x}_{\text{r}}^i; \boldsymbol{\theta}); \lambda]||_2^2. \tag{7}$$

Here $\boldsymbol{\theta}$ denotes the parameters of neural networks. The $\mathcal{L}_{\text{ic}}$ and $\mathcal{L}_{\text{bc}}$ measure the prediction error on initial training data $\{\boldsymbol{x}_{\text{ic}}^i\}_i^{N_{\text{ic}}}$ and boundary training data $\{t_{\text{bc}}^i, \boldsymbol{x}_{\text{bc}}^i\}_i^{N_{\text{bc}}}$. The residual loss $\mathcal{L}_{\text{r}}$ is imposed to make the neural network satisfy the PDE constraint on a set of collocation points $\{t_{\text{r}}^i, \boldsymbol{x}_{\text{r}}^i\}_i^{N_{\text{r}}}$. To compute the residual loss $\mathcal{L}_{\text{r}}$, partial derivatives of the neural network output with respect to $t$ and $\boldsymbol{x}$ can be obtained via automatic differentiation techniques (Baydin et al., 2018).

Despite the effectiveness, the introduction of physics-based loss function makes the optimization more ill-conditioned (Wang et al., 2021; Krishnapriyan et al., 2021). Efforts have been made to alleviate this problem from the aspects of loss weight balancing (Wang et al., 2021; 2022), loss function design (Psaros et al., 2022; Yu et al., 2022), adaptive collocation point sampling (Wight & Zhao, 2020; McClenny & Braga-Neto, 2020; Lu et al., 2021a; Wu et al., 2022), domain decomposition (Jagtap et al., 2020c; Jagtap & Karniadakis, 2020), and curriculum learning (Krishnapriyan et al., 2021). Different from previous works, we shed light on the relationship between the optimization difficulty of PDE constraint and activation functions. Our work reveals the high sensitivity of PINNs to the choice of activation function and proposes to reduce the optimization difficulty of PINNs by learning specialized activation functions for different PDEs. We hope our work could inspire further study on the convergence issue of PINNs from the perspective of activation functions.

## 2.2 ACTIVATION FUNCTIONS

**Fixed-shape activation function.** Activation functions are crucial for the optimization and performance of deep neural networks (Glorot & Bengio, 2010). The Rectified Linear Units (ReLU) (Hahnloser et al., 2000; Jarrett et al., 2009; Nair & Hinton, 2010) outperform the logistic and hyperbolic tangent activation functions and have become the default choice for most neural networks. Since then, designing activation functions has been an active research direction. Variants with improved learning characteristics are proposed, such as Softplus (Dugas et al., 2000; Glorot et al., 2011), Leaky ReLU (Maas et al., 2013), ELU (Clevert et al., 2015) and GELU (Hendrycks & Gimpel, 2016). Besides these hand-designed activation functions, Ramachandran et al. (2017) leverages neural architecture search techniques to discover the novel Swish function. In this work, we focus on the selection of activation functions for PINNs when solving different problems. While the common practice to find the optimal activation function is by trial-and-error, we formulate this manual selection as a learning problem, which can be optimized efficiently.

**Adaptive activation functions** are explored to find specialized activation functions for different architectures and tasks. The major difference of existing methods lies in the search space. The piece-wise linear function is adopted as the universal function approximator in some methods, such as APL (Agostinelli et al., 2014), RePLU (Li et al., 2016) and PWLU (Zhou et al., 2021). The formulation of SLAF (Goyal et al., 2019) is based on Taylor approximation with polynomial basis. PAU (Molina et al., 2019) leverages Padé approximation to form its search space. Motivated by the connection between Swish and ReLU, ACON (Ma et al., 2021) is proposed as a smooth approximator to the general Maxout family activation functions (Goodfellow et al., 2013). Our work proposes to learn an adaptive activation function as a weighted sum of candidate functions, whose weights can be adapted to the underlying physics laws when modelling different PDE systems. While similar ideas have been studied for convolutional neural networks in image classification (Dushkoff & Ptucha, 2016; Qian et al., 2018; Manessi & Rozza, 2018; Sütfeld et al., 2020), some technical challenges remain unexplored in the context of PINNs, which have a higher demand for the smoothness and diversity of the candidate functions. First, the optimization of PDE-based constraints needs the activation function to provide higher-order derivatives, which causes the failure of widely-used

ReLUs in PINNs. Second, unlike the image classification tasks, different PDE systems could have various characteristics, such as periodicity and rapid decay. This leads to a higher requirement for the diversity of the candidate functions. To overcome these challenges, we propose to build the candidate function set with simple elementary functions to embed the prior knowledge of physics systems, as well as commonly-used activation functions to ensure the diversity.

## 3 METHOD

In this section, we first investigate the influence of activation functions in PINNs for solving simple ODE systems with analytical solutions. The results show that the choice of activation functions is crucial for PINNs and depends on the problem. Motivated by this observation, we propose to learn specialized activation functions for different PDE systems.

### 3.1 ACTIVATION FUNCTIONS IN PINNS

Across the deep learning community, the ReLU enjoys widespread adoption and becomes the default activation function. However, there is no such default choice of activation functions for PINNs, regardless of the importance of activation functions in the optimization and expressivity of neural networks. One of the reasons lies in the failure of ReLUs in PINNs, whose second-order derivative is zero everywhere. More importantly, PINNs show great sensitivity to activation functions due to the various characteristics of the underlying PDE system. To give a further illustration, we compare the performance of PINNs with different activation functions on the following toy examples.

**Problem formulation.**    Here we consider the one-dimensional Poisson's equation as

$$\Delta u(x) = f(x),\ x \in [0, \pi], \tag{8}$$

with boundary condition $u(0) = 0$ and $u(\pi) = \pi$, where $\Delta$ is the Laplace operator and $f(x)$ denotes the source term. As shown in Table 1, we use different source terms to construct analytical solutions with various characteristics. For the problem **P1**, the solution is composed of trigonometric functions with different periods; for the problem **P2**, its solution is dominant by the exponential function. Following the setting described in Yu et al. (2022), we impose the boundary conditions as hard-constraints by choosing the surrogate of solution as

$$\hat{u}(x) = x(x - \pi)u'(x; \theta) + x, \tag{9}$$

where $u'(x; \theta)$ is the output of PINN. Hence, the training objective is reduced to the residual loss $\mathcal{L}_\mathrm{r}$. Note that we consider an extreme case of insufficient collocation points to highlight the difference between activation functions. The details of activation functions and experiment setups can refer to Appendix A and Appendix C, respectively.

Table 1: The source terms and analytical solutions for the 1D Poisson's equation.

| Problems | $f(x)$ | $u(x)$ |
|---|---|---|
| **P1** | $-(\sum_{i=1}^{20} i\sin(ix))$ | $x + \sum_{i=1}^{20} \sin(ix)/i$ |
| **P2** | $(x^3 - 2\pi x^2 + (\pi^2 + 3)x - 2\pi)e^{\frac{(x-\pi)^2}{2}}$ | $xe^{\frac{(x-\pi)^2}{2}}$ |

**PINNs exhibit high sensitivity to the choice of activation functions.**    Table 2 shows the results of different activation functions on **P1** and **P2**. As we can see, the choice of activation functions makes a big difference on the prediction accuracy. For example, there exists a large performance gap between the $\exp$ and $\tanh$ functions on **P2** ($0.73 \pm 2.08$ vs. $75.77 \pm 7.93$). Moreover, the performance of the same activation function can vary significantly across different problems. One can find that while $\sin$ achieves the lowest error on **P1**, it produces poor results on **P2**. A similar phenomenon is observed on the $\exp$ function. Note that none of these activation functions can achieve the best performance simultaneously on these two problems.

**We argue that this sensitivity arises from the various characteristics of each problem.**    In Figure 1, we visualize the second-order derivative of PINNs to analyse the influence of activation functions on the optimization of residual loss $\mathcal{L}_\mathrm{r}$. We find that the commonly-used $\tanh$ function

fails to model the periodic nature of **P1** and the rapid decay of **P2**, exhibiting severe overfitting with insufficient collocation points. In contrast, accurate approximation can be achieved under the same conditions by using activation functions with suitable properties, such as the $\sin$ function for **P1** and the $\exp$ function for **P2**. However, there dose not exist a generic activation function which could model different characteristics simultaneously. This finding demonstrates the necessity of careful selection of activation functions when solving different problems.

Table 2: Comparisons between different activation functions. $L_2$ relative error (%) is reported.

| Problems | sin | exp | tanh | sigmoid | Softplus | ELU | GELU | Swish |
|---|---|---|---|---|---|---|---|---|
| **P1** | **0.91 ± 0.30** | 39.11 ± 31.32 | 34.27 ± 45.21 | 23.51 ± 17.82 | 86.03 ± 64.79 | 36.68 ± 20.61 | 19.48 ± 18.50 | 33.16 ± 36.81 |
| **P2** | 49.42 ± 4.75 | **0.73 ± 2.08** | 75.77 ± 7.93 | 56.36 ± 10.81 | **0.65 ± 0.23** | 40.96 ± 97.75 | 2.70 ± 3.41 | 1.05 ± 0.42 |

(a) The second-order derivative of PINNs for **P1**.     (b) The second-order derivative of PINNs for **P2**.

Figure 1: Visualization of the residual loss $\mathcal{L}_r$. In the case of 1D Poisson's equation, the residual loss $\mathcal{L}_r$ penalizes the deviation of the second-order derivative of PINNs to the exact values on sampled collocation points (black dots in figures). We plot the results of $\sin$, $\exp$ and $\tanh$.

## 3.2 LEARNING SPECIALIZED ACTIVATION FUNCTIONS FOR SOLVING PDEs

In light of the diversity and complexity of PDEs, it is critical to select proper activation functions in PINNs. Existing works usually make the choice though trial-and-error. However, this strategy is inefficient and leads to the impractical combinatorial search problem in some cases where choosing a set of suitable activation functions is necessary for solving complex problems. To address this problem, we propose a simple and effective method to search for the optimal activation functions for different problems automatically.

To be specific, we formulate the manual selection of activation functions as a learning problem, following the spirits of neural architecture search. Our basic idea is to construct the search space with a set of candidate activation functions and learn to predict the optimal activation function for different PDEs. Instead of categorically selecting one particular activation function, we relax the search space to be a linear combination of all candidate activation functions with learnable coefficients. This continuous search space allows efficient optimization through backpropagation and enables the discovery of novel activation functions.

Formally, we define the physics-informed activation function (PIAC) as

$$\text{PIAC}(x) = \sum_{i=1}^{N} G(\alpha_i)\sigma_i(x), \; G(\alpha_i) = \frac{\exp(\alpha_i)}{\sum_{j=1}^{N} \exp(\alpha_j)} \qquad (10)$$

where $\sigma_i(\cdot)$ and $\alpha_i$ denote a candidate activation function and a learnable parameter, respectively. Together with a gate function $G(\cdot)$, the parameter $\alpha_i$ determines the weight (or coefficient) of its corresponding candidate activation function. We use the softmax as the gate function by default. In this case, the search space is the convex hull of the set of candidate activation functions. When the weights are fixed as one-hot vectors, the proposed PIAC is reduced to its candidate function. When the weights are learnable, a specialized novel activation function can be efficiently searched for different PDE systems thanks to the continuous parameterization. We employ the PIAC by replacing all the activation functions in PINNs. It can be applied in a neuron-wise manner where learnable parameters are allowed to be vary across neurons, or in a layer-wise manner where the parameters are shared for all neurons in one layer.

The search space of PIAC is built upon the set of candidate functions $\mathcal{F} = \{\sigma_1, \sigma_2, ..., \sigma_N\}$. Conceptually, we build this candidate function set with simple elementary functions and commonly-used activation functions. First, we can embed prior knowledge about the PDE system into the networks by including appropriate elementary functions. For example, the periodicity and exponential decay are commonly seen in physics equations but rarely observed in image data. As shown in Section 3.1, they are difficult to model by neural networks with normal activation functions such as $\tanh$. We can alleviate the modeling difficulty by taking advantage of sinusoidal and exponential functions in our search space. We can also add those elementary functions observed in the initial condition (see the convection equation) or the force term (see Poisson's equation) to help the optimization of corresponding loss function. Second, we add most of commonly-used activation functions into this function set. By doing this, we ensure the diversity of our search space while avoiding the repetitive evaluations of each candidate activation function for different problems. Moreover, the search space can be further enhanced with the adaptive slope, which is proposed to improve the convergence rate of standard activation functions for PINNs (Jagtap et al., 2020b;a). This can be achieved by simply introducing a learnable scaling factor $\beta_i$ for each candidate function $\sigma_i$, which can be represented as

$$\text{PIAC}(x) = \sum_{i=1}^{N} G(\alpha_i)\sigma_i(\beta_i x). \tag{11}$$

## 4 EXPERIMENTS

In this section, We evaluate the performance of our method on several benchmarks of time-dependent PDEs. Then, we provide detailed ablation results to analyse the proposed PIAC. Finally, we present an intuitive interpretation of the performance gain of our method from the neural tangent kernel perspective. More results on the 1D Poisson's equation can be found in Appendix C.

### 4.1 EXPERIMENTS ON TIME-DEPENDENT PDES

We demonstrate the generalization ability of PIAC to solve various time-dependent PDEs, ranging from first-order linear PDE to fourth-order nonlinear PDE, including the convection equation, the Burgers' equation, the Allen-Cahn equation, the Korteweg–de Vries equation, and the Cahn-Hilliard equation. Note that for the linear convection problem, we consider the case of a high convection coefficient ($\beta$=64), which is shown as a difficult problem for vanilla PINNs (Krishnapriyan et al., 2021). The details of PDEs and experimental configurations can refer to Appendix D.

**Baselines and PIAC setups.** We compare PIAC with several commonly-used activation functions, including the sinusoidal functions (sin), the hyperbolic tangent function ($\tanh$), the logistic function (sigmoid), the Softplus function (Softplus), the Exponential Linear Unit (ELU), the Gaussian Error Linear Unit (GELU) and the Swish function (Swish). We compare PIAC to other adaptive activation functions which could provide higher-order derivatives, including SLAF Goyal et al. (2019), PAU Molina et al. (2019) and ACON Ma et al. (2021). The details of each activation function can refer to Appendix A and Appendix B. We also compare PIAC with standard activation functions with the layer-wise adaptive slopes (Jagtap et al., 2020a). We employ the PIAC in a layer-wise manner by default. We set the candidate function set $\mathcal{F}$ as $\{\sin, \tanh, \text{GELU}, \text{Swish}, \text{Softplus}\}$. The learnable parameters $\{\alpha_i\}_{i=1}^{N}$ are initialized as zeros and optimized jointly with the weights and biases of PINNs. The scaling factors $\{\beta_i\}_{i=1}^{N}$ are initialized as ones and can be fixed or learnable.

#### 4.1.1 MAIN RESULTS

**Comparisons with fixed activation functions.** Table 3 presents the $L_2$ relative error of PIAC and seven standard activation functions on five time-dependent PDEs. One can observe that PIAC outperforms standard activation functions on all PDEs. For example, PIAC reduces the error rate by 83% on the convection equation ($0.06 \pm 0.03\%$ vs. $0.36 \pm 0.15\%$), by 46% on the KdV equation ($0.34 \pm 0.05\%$ vs. $0.64 \pm 0.34\%$), and by 56% on the Cahn-Hilliard equation ($0.50 \pm 0.15\%$ vs. $1.01 \pm 1.27\%$), compared with optimal standard activation function of each problem. More importantly, PIAC performs consistently over all these problems, while standard activation functions suffer from high variance in the performance of different PDEs.

Table 3: Comparisons of standard activation functions,PIAC and their counterparts with adaptive slopes (AS) on time-dependent PDEs. $L_2$ relative error (%) is reported. We also report the average error rate over all problems. The better results are **bold-faced**.

| Method | Convection equation | Burgers' equation | Allen-Cahn equation | KdV equation | Cahn-Hilliard equation | Average error |
|---|---|---|---|---|---|---|
| sin | $0.36 \pm 0.15$ | $5.18 \pm 3.73$ | $3.57 \pm 0.64$ | $0.63 \pm 0.17$ | $1.72 \pm 0.61$ | 2.29 |
| tanh | $6.83 \pm 4.79$ | $0.26 \pm 0.13$ | $1.34 \pm 0.54$ | $1.32 \pm 1.12$ | $4.02 \pm 4.56$ | 2.75 |
| sigmoid | $70.38 \pm 2.98$ | $1.24 \pm 1.05$ | $1.63 \pm 0.13$ | $2.34 \pm 0.53$ | $3.12 \pm 2.72$ | 15.74 |
| GELU | $39.29 \pm 35.51$ | $4.43 \pm 2.87$ | $3.93 \pm 0.78$ | $1.21 \pm 0.41$ | $1.01 \pm 1.27$ | 9.97 |
| Swish | $5.59 \pm 2.18$ | $8.27 \pm 4.35$ | $5.56 \pm 1.28$ | $1.73 \pm 0.10$ | $2.22 \pm 2.60$ | 4.67 |
| Softplus | $55.39 \pm 2.46$ | $17.75 \pm 8.11$ | $17.72 \pm 6.04$ | $6.23 \pm 0.44$ | $9.67 \pm 2.57$ | 21.35 |
| ELU | $6.67 \pm 0.96$ | $46.34 \pm 2.36$ | $52.55 \pm 2.95$ | $78.95 \pm 2.57$ | $90.77 \pm 2.07$ | 55.66 |
| SLAF | $0.36 \pm 0.18$ | $43.93 \pm 0.66$ | $33.93 \pm 9.97$ | $25.23 \pm 1.28$ | $52.57 \pm 27.93$ | 31.20 |
| PAU | $45.78 \pm 35.47$ | $48.31 \pm 8.21$ | $43.83 \pm 15.18$ | $68.11 \pm 13.49$ | $115.59 \pm 3.79$ | 64.32 |
| ACON | $3.55 \pm 1.66$ | $1.18 \pm 1.55$ | $3.88 \pm 1.82$ | $1.52 \pm 0.35$ | $2.46 \pm 1.96$ | 2.52 |
| **PIAC** | $\mathbf{0.06 \pm 0.03}$ | $\mathbf{0.21 \pm 0.11}$ | $\mathbf{0.76 \pm 0.26}$ | $\mathbf{0.34 \pm 0.05}$ | $\mathbf{0.50 \pm 0.15}$ | **0.37** |
| With AS | | | | | | |
| sin | $0.28 \pm 0.08$ | $1.82 \pm 1.58$ | $3.57 \pm 0.46$ | $0.57 \pm 0.19$ | $1.73 \pm 0.73$ | 1.59 |
| tanh | $3.06 \pm 1.65$ | $0.16 \pm 0.09$ | $0.81 \pm 0.21$ | $1.71 \pm 1.53$ | $2.11 \pm 0.49$ | 1.57 |
| GELU | $38.00 \pm 39.53$ | $0.71 \pm 0.38$ | $1.99 \pm 0.63$ | $0.79 \pm 0.17$ | $0.96 \pm 0.57$ | 8.49 |
| **PIAC** | $\mathbf{0.05 \pm 0.02}$ | $\mathbf{0.15 \pm 0.10}$ | $\mathbf{0.58 \pm 0.15}$ | $\mathbf{0.34 \pm 0.08}$ | $\mathbf{0.35 \pm 0.07}$ | **0.29** |

**Comparisons with other adaptive activation functions.** The results of different adaptive activation functions are shown in Table 3. SLAF achieves performance comparable to the best standard activation function on the first-order convection equation, but does not produce accurate predictions for other higher-order PDEs. Although SLAF shares a similar formulation with PIAC, its polynomial bases can cause vanishing or exploding gradients due to the high order powers in its derivatives. This problem might hinder the optimization of higher-order PDE-based constraints. PAU suffers from instability of training (see Appendix Figure 2) and performs poorly on all PDEs. We attribute the training instability of PAU to its discontinuous derivatives, which arise from the absolute value in the denominator of its formulation. ACON achieves stable performance but does not surpass the best standard activation function in each problem due to the limited flexibility of its formulation. The proposed PIAC outperforms these methods thanks to the diverse and smooth candidate functions, which can be used to incorporate the prior knowledge of different physics systems.

**Comparisons with activation functions with adaptive slopes.** We also compare PIAC with three standard activation function enhanced with adaptive slopes (AS). We find PIAC without AS achieves competitive or even better results. For example, PIAC without AS outperforms the tanh with AS on the Allen-Cahn equation and outperforms the sin with AS on the convection equation. The performance of PIAC can be further improved by incorporating adaptive slopes into the search space. As a showcase, PIAC with AS obtains a performance gain by 64% compared to the GELU with AS on the Cahn-Hilliard equation and by 82% compared to the sin with AS on the convection equation. Furthermore, our method has a better generalization ability and achieves the lowest average error rate over all problems (0.29% with AS and 0.37% without AS), while the adaptive slope technique cannot alleviate the need of the manual selection of activation functions.

**Visualization of the learned PIAC.** We show the learned PIAC in Appendix Figure 3. One can observe the learned activation functions are different from candidate functions. We also find differences in learned functions of different layers. For example, a deeper layer tends to has a larger weight of sin function in the cases of convection equation. Moreover, the learned activation functions vary across problems, which conforms to our motivation to learn specialized activation functions for different PDE systems. The prediction results of PIAC for all problems can refer to Appendix Figure 4.

## 4.2 ABLATION STUDY

We ablate different design choices to provide a better understanding of the proposed method. For all ablation experiments, we deploy layer-wise PIAC to solve the convection equation. The adaptive slope is deactivated for a clear comparison.

Table 4: PIAC ablation experiments on the convection equation. We repeat each experiments 5 times and report the average $L_2$ relative error (%). Default settings are marked in gray .

(a) Comprisons of different candidate function set.

| Function set $\mathcal{F}$ | #params | error (%) |
|---|---|---|
| {sin, tanh, GELU, Swish, Softplus} | 16922 | **0.06 ± 0.03** |
| {sin, tanh, GELU, Swish} | 16917 | **0.06 ± 0.03** |
| {sin, tanh, GELU} | 16912 | 0.10 ± 0.12 |
| {sin, tanh} | 16907 | 0.33 ± 0.04 |
| sin | 16897 | 0.36 ± 0.15 |
| tanh | 16897 | 6.83 ± 4.79 |
| GELU | 16897 | 39.29 ± 35.51 |

(b) Comprisons of different search spaces.

| $G(\cdot)$ | Init | Learnable | error (%) |
|---|---|---|---|
| identity | 1/N | ✓ | 0.22 ± 0.22 |
| $L_1$-norm | 1/N | ✓ | 1.18 ± 0.93 |
| sigmoid | Zero | ✓ | 0.89 ± 0.83 |
| softmax | Zero | ✓ | **0.06 ± 0.03** |
| softmax | Zero | ✗ | 1.60 ± 1.12 |
| softmax | Random | ✗ | 0.32 ± 0.23 |
| softmax | Random | ✓ | 0.09 ± 0.05 |

**Comparing different candidate function set.**   Here we study the influence of candidate function set by gradually increasing its size as shown in Table 4a. We begin with the two most commonly-used activation functions, the sin and tanh functions. In this case, PIAC is shown to achieve competitive result compared with its two candidate functions. Based on that, we observe that adding new activation functions could lead to a better performance. Surprisingly, the introduction of GELU can still improve the performance despite its poor results on this problem. We argue that this performance gain arises from the enlarged search space. We notice that the performance tends to saturate as the number of candidate functions increases. This indicates that PIAC has a good robustness to the choice of candidate function set if this set contains activation functions with sufficient diversity. Notably, the additional parameters of PIAC is negligible compared with total number of weights.

**Comparing different gate functions.**   We also compare the softmax function with other gate functions in Table 4b. The identity function $G(\alpha_i) = \alpha_i$ includes all linear combinations of candidate activation functions into the search space; while the sigmoid function $G(\alpha_i) = 1/(1 + \exp(-\alpha_i))$ restricts the coefficients to be between 0 and 1. We also consider the $L_1$-normalization $G(\alpha_i) = \alpha_i / \sum_{j=1}^{N} |\alpha_j|$, which allows for negative coefficients and keeps the competition between candidate functions. The trainable $\{\alpha_i\}_{i=1}^{N}$ are initialized as zeros for sigmoid and softmax functions and as $1/N$ for identity and $L_1$-normalization functions, where $N$ denotes the number of candidate functions. One can observe that the results of softmax function are more stable and more accurate. This implies proper restrictions of the search space can work as a regularization to the learning of PIAC.

**Effectiveness of learnable coefficients of PIAC.**   To demonstrate that the learnable coefficients of PIAC is effective and necessary, we compare PIAC to two variants with fixed coefficients. The first one initializes its parameters as zeros and the second one initializes its parameters from a standard normal distribution. As shown in Table 4b, the automatically learned coefficients outperforms evenly distributed coefficients and random sampled coefficients by a large margin. Furthermore, one can observe that the performance gap between the random and constant initialization is eliminated if these parameters are trainable, which also demonstrates the effectiveness of PIAC's optimization.

## 4.3   UNDERSTANDING PIAC THROUGH THE LENS OF NEURAL TANGENT KERNEL

The neural tangent kernel is proposed to describe the evolution of neural networks (Jacot et al., 2018). Its eigenvalues can be leveraged to analyse the rate of convergence as shown in previous works (Jacot et al., 2018; Tancik et al., 2020; Wang et al., 2022). We show that the introduction of PIAC makes the NTK learnable. Empirically, we observe the optimization of PIAC adapts the NTK's eigenvalue spectrum to the underlying PDE system and leads to an improvement on the average eigenvalue, which partially explains the stable and fast convergence of PIAC.

**Brief introduction of NTK.**   Specially, the neural tangent kernel of a $L$-layer fully-connected network $u^{(L)}$ with parameters $\boldsymbol{\theta}$ is defined as

$$\Theta^{(L)}(\boldsymbol{x}, \boldsymbol{x}') = \langle \frac{\partial u^{(L)}(\boldsymbol{x}|\boldsymbol{\theta})}{\partial \boldsymbol{\theta}}, \frac{\partial u^{(L)}(\boldsymbol{x}'|\boldsymbol{\theta})}{\partial \boldsymbol{\theta}} \rangle = \sum_{\theta_i} \frac{\partial u^{(L)}(\boldsymbol{x}|\boldsymbol{\theta})}{\partial \theta_i} \frac{\partial u^{(L)}(\boldsymbol{x}|\boldsymbol{\theta})}{\partial \theta_i} \qquad (12)$$

where $\boldsymbol{x}$ and $\boldsymbol{x}'$ are two inputs. As demonstrated in Jacot et al. (2018), in the infinite-width limit and under suitable conditions, the NTK $\Theta^{(L)}$ converges to a deterministic kernel $\Theta_\infty^{(L)}$ at initialization and stays constant during the training. This limiting kernel can be represented as

$$\Theta_\infty^{(L)} = \sum_{l=1}^{L} \left( \Sigma^{(l)}(\boldsymbol{x}, \boldsymbol{x}') \prod_{l'=l+1}^{L} \dot{\Sigma}^{(l')}(\boldsymbol{x}, \boldsymbol{x}') \right),$$

$$\Sigma^{(l+1)}(\boldsymbol{x}, \boldsymbol{x}') = \mathbb{E}_{f \sim \mathcal{N}(0, \Sigma^{(l)})}[\sigma(f(\boldsymbol{x}))\sigma(f(\boldsymbol{x}'))] + \beta^2,$$

$$\dot{\Sigma}^{(l+1)}(\boldsymbol{x}, \boldsymbol{x}') = \mathbb{E}_{f \sim \mathcal{N}(0, \Sigma^{(l)})}[\dot{\sigma}(f(\boldsymbol{x}))\dot{\sigma}(f(\boldsymbol{x}'))] + \beta^2.$$

(13)

Note $\Sigma^{(1)}(\boldsymbol{x}, \boldsymbol{x}') = \frac{1}{n_0}\boldsymbol{x}^T\boldsymbol{x}' + \beta^2$. The expectation in Eq.(13)is taken with respect to a centered Gaussian process $f$ with covariance $\Sigma^{(l)}$; $n_0$ is the dimension of input $\boldsymbol{x}$ and $\beta$ is a scaling factor; $\sigma$ and $\dot{\sigma}$ denote the activation function and its derivative, respectively. Note that the covariance $\Sigma^{(l)}$ depends on the choice of activation functions and so dose the NTK $\Theta_\infty^{(L)}$.

**PIAC and the learnable NTK.**  For simplicity, we derive the NTK of standard neural networks with PIAC. The NTK of PINNs with PIAC can be derived in a similar way. Firstly, we assume that the optimization of neural networks with PIAC can be decomposed into two phases, where we learn the coefficients of PIAC in the first phase and then train the parameters of neural network in the second phase. This assumption is reasonable as the number of parameters of PIAC is far less than those of networks and they quickly converge at the early stage of training. Empirically, we observe that a short warmup (2.5% of the whole schedule) is sufficient for PIAC to learn suitable activation functions and to achieve competitive performance compared with the counterpart whose coefficients are updated during the whole schedule (0.58% vs. 0.60% on the Allen-Cahn equation). By decoupling the updates of PIAC and networks, we find the second optimization phase is equivalent to the training of a standard network with a learned activation function, whose NTK is derived as

$$\bar{\Theta}_\infty^{(L)} = \sum_{l=1}^{L} \left( (\sum_{i=1}^{N}\sum_{j=1}^{N} G(\alpha_i)G(\alpha_j)\Sigma_{\sigma_i\sigma_j}^{(l)}(\boldsymbol{x}, \boldsymbol{x}')) \prod_{l'=l+1}^{L} (\sum_{i=1}^{N}\sum_{j=1}^{N} G(\alpha_i)G(\alpha_j)\dot{\Sigma}_{\sigma_i\sigma_j}^{(l')}(\boldsymbol{x}, \boldsymbol{x}')) \right),$$

$$\Sigma_{\sigma_i\sigma_j}^{(l+1)}(\boldsymbol{x}, \boldsymbol{x}') = \mathbb{E}_{f \sim \mathcal{N}(0, \Sigma^{(l)})}[\sigma_i(f(\boldsymbol{x}))\sigma_j(f(\boldsymbol{x}'))],$$

$$\dot{\Sigma}_{\sigma_i\sigma_j}^{(l+1)}(\boldsymbol{x}, \boldsymbol{x}') = \mathbb{E}_{f \sim \mathcal{N}(0, \Sigma^{(l)})}[\dot{\sigma}_i(f(\boldsymbol{x}))\dot{\sigma}_j(f(\boldsymbol{x}'))].$$

(14)

We omit the factor $\beta$ for convenience. One can observe that the covariance $\Sigma^{(l)}$ in Eq(13) is replaced with a weighted sum of $\Sigma_{\sigma_i\sigma_j}^{(l)}$, which is calculated with different combinations of candidate activation functions. Note that the weights are learned in the first training phase. To conclude, the introduction of PIAC leads to a learnable NTK which could be adapted to the underlying PDE.

**The eigenvalue spectrum of NTK with PIAC.**  Through the optimization of PIAC, we can modify the limiting NTK $\bar{\Theta}_\infty^{(L)}$, which corresponds to a change in the convergence behavior of the neural network. We show empirically that the optimization of PIAC has an effect on the NTK's eigenvalue spectrum. As shown in Appendix Figure 3d, PIAC leads to a larger average eigenvalue compared with the best standard activation function on the Allen-Cahn equation (5859 vs. 2407), which implies a larger convergence rate (Wang et al., 2022).

## 5  CONCLUSION

In this paper, we reveal the high sensitivity of PINNs to the choice of activation functions and relate it to various characteristics of the underlying PDE system. We sought to learn specialized activation functions automatically for PINNs to avoid the inefficient manual selection of activation functions and to alleviate the optimization difficulty of PINNs. The proposed physics-informed activation function is presented as learnable combinations of a set of candidate functions, whose coefficients can be adapted to the governing PDEs. Intuitively, we show that PIAC makes the neural tangent kernel of PINNs learnable, which partially explains the performance improvement of PIAC. Extensive experiments on a series of challenging benchmarks demonstrate the effectiveness and generalization ability of PIAC.

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

## A  STANDARD ACTIVATION FUNCTIONS

We compare the proposed PIAC with several standard activation functions for solving a series of PDEs. The details are shown as follows.

- sin:
$$f(x) = \sin(\beta * x), \tag{15}$$
  where scaling factor $\beta$ is set to 1 as default.

- exp:
$$f(x) = e^{(\beta * x)} - 1, \tag{16}$$
  where $\beta$ is set to 1 as default. For the **P1** setting of 1D Poisson's equation, $\beta$ is set to 0.25 for better performance.

- tanh:
$$f(x) = \frac{e^x - e^{-x}}{e^x + e^{-x}}. \tag{17}$$

- sigmoid:
$$f(x) = \frac{1}{1 + e^{-x}}. \tag{18}$$

- Softplus (Dugas et al., 2000; Glorot et al., 2011):
$$f(x) = \frac{1}{\beta} \log(1 + e^{\beta * x}), \tag{19}$$
  where $\beta = 1$. Softplus can be regard as a smooth version of ReLU.

- the Exponential Linear Unit (ELU) (Clevert et al., 2015):
$$f(x) = \begin{cases} x, & \text{if } x \geq 0, \\ e^x - 1, & \text{if } x < 0. \end{cases} \tag{20}$$

- the Gaussian Error Linear Unit (GELU) (Hendrycks & Gimpel, 2016):
$$f(x) = x * \Phi(x), \tag{21}$$
  where $\Phi(x)$ is the cumulative distribution function of the standard Gaussian distribution.

- Swish (Hendrycks & Gimpel, 2016; Ramachandran et al., 2017; Elfwing et al., 2018):
$$f(x) = \frac{x}{1 + e^{-x}}. \tag{22}$$

## B  ADAPTIVE ACTIVATION FUNCTIONS

We further compare the proposed PIAC with other adaptive activation functions which could provide higher-order derivatives. The details are shown as follows.

- The SLAF Goyal et al. (2019) proposes to learn activation functions with Taylor polynomial bases, which is presented as:
$$f(x) = \sum_{i=0}^{m} a_i x^i, \tag{23}$$
  where m is set to 5 in our experiments. The learnable parameters $\{a_i\}_{i=0}^{m}$ are initialized from a normal distribution.

- The PAU Molina et al. (2019) is presented as the rational functions of the form
$$f(x) = \frac{\sum_{i=0}^{m} a_i x^i}{1 + |\sum_{i=1}^{n} b_i x^i|}, \tag{24}$$
  where m is set to 5 and n is set to 4 following the default setting in Molina et al. (2019). The absolute value is to avoid a zero-valued denominator. The learnable parameters $\{a_i\}_{i=0}^{m}$ and $\{b_i\}_{i=1}^{n}$ are initialized from a normal distribution.

- The ACON Ma et al. (2021) is a smooth approximator to the general Maxout family activation functions, which is presented as
$$f(x) = (p_1 - p_2)x * \text{sigmoid}(\beta(p_1 - p_2)x) + p_2 x, \tag{25}$$
  where $p_1$, $p_2$ and $\beta$ are learnable parameters and initialized from a normal distribution.

## C  EXPERIMENTS ON 1D POISSON'S EQUATION

**Experiment setups.**  For each problem, we use a 4-layer multilayer perceptron (MLP) with the hidden dimension set to 16. The model is initialized by the Xavier initialization (Glorot & Bengio, 2010) and trained for 20000 iterations using the Adam optimizer (Kingma & Ba, 2014). We initialize the learning rate as 1e-3 and adapt it with a half-cycle cosine decay schedule. We use 32 collocation points for both **P1** and **P2**. For each problem, we experiment with 8 standard activation functions as described in Appendix A. We repeat each experiment 10 times and report the mean and standard deviation of the $L_2$ relative error.

**Comparisons between PIAC and standard activation functions.**  To demonstrate the effectiveness of PIAC, we use it to learn combinations of the $\sin$ and $\exp$ functions for solving the 1D Poisson's equation as described in Section 3.1. As shown in Table 2, although the $\sin$ and $\exp$ functions can achieve the lowest $L_2$ relative error on **P1** and **P2** respectively, they perform poorly on the other problem. We expect the PIAC could adapt its attention over these two candidate functions according to the problem. We define $\mathcal{F} = \{\sin, \exp\}$. The trainable parameters $\{\alpha_i\}_{i=1}^N$ are initialized as zeros, which implies equal attention over all candidate functions.

The results of PIAC and its two candidate functions are presented in Table 5a. It can be observed that PIAC can achieve comparable accuracy with the $\sin$ on **P1** and outperform the $\exp$ on **P2**, despite the large performance gap between its two candidate functions. We also report the values of learned coefficients in Table 5b. For **P1**, PIAC pays more attention on the $\sin$ function as expected. For **P2**, we observe that the learned activation functions are different across layers. The $\exp$ is preferred only on the third layer, which still leads a performance gain compared with standard activation functions. Overall, the results indicate that the proposed method can learn useful combination coefficients by considering the effectiveness of candidate functions for different problems.

Table 5: Comparisons of standard activation and PIAC on the 1D Poisson's equation. $L_2$ relative error (%) is reported. We also report the learned coefficients of $\sin$, $\alpha_{\sin}$. Note that the coefficients of $\exp$ can be obtained by $\alpha_{\exp} = 1 - \alpha_{\sin}$, since the softmax gate function is used.

(a) The $L_2$ relative error (%).

| Problem | **P1** | **P2** |
|---|---|---|
| sin | $0.91 \pm 0.30$ | $49.42 \pm 4.75$ |
| exp | $39.11 \pm 32.32$ | $0.73 \pm 2.08$ |
| PIAC | $\mathbf{0.88 \pm 0.42}$ | $\mathbf{0.01 \pm 0.006}$ |

(b) The learned coefficients $\alpha_{\sin}$ of different layers.

| Layers | **P1** | **P2** |
|---|---|---|
| 1 | $0.81 \pm 0.02$ | $0.58 \pm 0.08$ |
| 2 | $0.75 \pm 0.06$ | $0.53 \pm 0.05$ |
| 3 | $0.84 \pm 0.04$ | $0.45 \pm 0.06$ |

## D  EXPERIMENTS ON TIME-DEPENDENT PDES

### D.1  PROBLEM FORMULATIONS

**Convection equation.**  Consider a one-dimensional linear convection problem as

$$u_t + \beta u_x = 0, \; x \in [0, 2\pi], \; t \in [0, 1],$$
$$u(0, x) = \sin(x), \; u(t, 0) = u(t, 2\pi). \tag{26}$$

The solution of this problem is periodic over time, whose period is inversely proportional to the convection coefficient $\beta$. Previous work (Krishnapriyan et al., 2021) finds it difficult for vanilla PINNs to learn the solution with a large $\beta$ and proposes to train the PINNs with curriculum learning. To be specific, the $\beta$ is initialized as a small value and gradually increases during the training. Despite its effectiveness, the training cost is increased significantly. We show that accurate prediction can be achieved with a normal training strategy by simply using a suitable activation function. Moreover, the performance can be further improved by our method. Here we set the convection coefficient $\beta$ to 64. We use $N_{\text{ic}} = 512$ training points for the initial condition and $N_{\text{bc}} = 200$ for the boundary condition. The residual loss $\mathcal{L}_r$ is computed on $N_r = 6400$ randomly sampled collocation points.

**Burgers' equation.** We consider the one-dimensional Burgers' equation:

$$u_t + uu_x = \nu u_{xx}, \ x \in [-1, 1], \ t \in [0, 1],$$
$$u(0, x) = -\sin(\pi x), \ u(t, -1) = u(t, 1) = 0, \tag{27}$$

where $\nu = 0.01/\pi$. Following (Lu et al., 2021a), we uniformly sample $N_{\mathrm{ic}} = 256$ and $N_{\mathrm{bc}} = 100$ points for initial and boundary training data, respectively. We compute the residual loss $\mathcal{L}_{\mathrm{r}}$ on $N_{\mathrm{r}} = 4800$ collocation points, which are randomly selected from the spatial-temporal domain.

**Allen-Cahn equation.** We next consider the Allen-Cahn equation as

$$u_t = Du_{xx} + 5(u - u^3), \ x \in [-1, 1], \ t \in [0, 1],$$
$$u(0, x) = x^2\cos(\pi x), \ u(t, -1) = u(t, 1) = -1, \tag{28}$$

where $D = 0.001$ following the setting in (Yu et al., 2022). We choose the following surrogate of solution to enforce the initial and boundary conditions:

$$\hat{u}(t, x) = x^2\cos(\pi x) + t(1 - x^2)u'(t, x; \theta), \tag{29}$$

where $u'(t, x; \theta)$ is the output of neural networks. We use $N_{\mathrm{r}} = 8000$ collocation points.

**Korteweg–de Vries equation.** We consider the Korteweg–de Vries (KdV) equation as

$$u_t + \lambda_1 uu_x + \lambda_2 u_{xxx} = 0, \ x \in [-1, 1], \ t \in [0, 1],$$
$$u(0, x) = \cos(\pi x), \ u(t, -1) = u(t, 1), \ u_x(t, -1) = u_x(t, 1), \tag{30}$$

where $\lambda_1 = 1$ and $\lambda_2 = 0.0025$ following the setting in (Raissi et al., 2019a). We choose the following surrogate of solution to enforce the initial conditions:

$$\hat{u}(t, x) = \cos(\pi x) + tu'(t, x; \theta), \tag{31}$$

where $u'(t, x; \theta)$ is the output of neural networks. We use $N_{\mathrm{bc}} = 200$ training points for the boundary condition and $N_{\mathrm{r}} = 8000$ collocation points for the residual loss $\mathcal{L}_{\mathrm{r}}$.

**Cahn-Hilliard equation.** We consider the phase space representation of Cahn-Hilliard equation as

$$u_t - \nabla^2(-\lambda_1\lambda_2 h + \lambda_2(u^3 - u)) = 0, \ h = \nabla^2 u, \ x \in [-1, 1], \ t \in [0, 1],$$
$$u(0, x) = \cos(\pi x) - \exp(-4(\pi x)^2), u(t, -1) = u(t, 1), \ u_x(t, -1) = u_x(t, 1), \tag{32}$$
$$h(t, -1) = h(t, 1), \ h_x(t, -1) = h_x(t, 1),$$

where $\lambda_1 = 0.02$ and $\lambda_2 = 1$ following the setting in (Mattey & Ghosh, 2022). Previous work proposes a novel sequential training method to solve this strongly non-linear and high-order problem (Mattey & Ghosh, 2022). We find that a normal training strategy can lead to a comparable result if we add more collocation points around $t = 0$. To be specific, we sample 4000 collocation points from time interval $[0, 0.05)$ and 8000 points from $[0.05, 1]$. We set $N_{\mathrm{ic}} = 256$ and $N_{\mathrm{bc}} = 100$. We show that PINNs under this training setting can work as a strong baseline and can be further improved by PIAC.

## D.2 EXPERIMENTAL CONFIGURATIONS

**Experiment setups.** We train the PINNs by a two-stage optimization except the convection equation. At the first stage, the model is trained for a certain number of iterations by Adam (Kingma & Ba, 2014). Then, the L-BFGS (Byrd et al., 1995) is used to train the network until convergence. The first stage is to provide a good initialization for the L-BFGS optimizer. In the case of convection equation, only the Adam optimizer is used. The detailed experimental configurations can be found in Table 6. The model is initialized by the Xavier initialization. The learning rate is adapted with a half-cycle cosine decay schedule in the first stage with Adam optimizer. We repeat each experiment 5 times and report the mean and standard deviation of the $L_2$ relative error. We run all experiments on one NVIDIA GeForce RTX 2080Ti GPU.

Table 6: The experimental configurations for all problems.

| | Network | | Adam | | L-BFGS | Data | | |
| Problems | Depth | Hidden dims | Iters | lr | Max Iters | $N_{\text{ic}}$ | $N_{\text{bc}}$ | $N_{\text{r}}$ |
| --- | --- | --- | --- | --- | --- | --- | --- | --- |
| Convection equation | 6 | 64 | 100k | 2e-3 | - | 512 | 200 | 6400 |
| Burgers' equation | 4 | 32 | 15k | 1e-3 | 15k | 256 | 100 | 5300 |
| Allen-Cahn equation | 4 | 32 | 40k | 1e-3 | 15k | - | - | 8000 |
| KdV equation | 4 | 32 | 40k | 1e-3 | 15k | - | 200 | 8000 |
| Cahn-Hilliard equation | 4 | 32 | 100k | 1e-3 | 15k | 256 | 100 | 15000 |

## D.3 VISUALIZATION

We show the training losses of different activation functions in Figure 2. One can observe SLAF seems to get stuck in local minima when solving the Burgers' equation and the Allen-Cahn equation. We find PAU suffers from instability of training as its loss oscillates severely. ACON achieves better performance than SLAF and PAU, but does not surpass tanh on both PDEs. Compared to these learnable activation functions, the training loss of PIAC is more stable and converges faster.

We present the learned activation functions of PIAC in Figure 3 and the prediction results of PIAC in Figure 4. We also visualize the eigenvalue spectrum of NTK with PIAC in Figure 3d, which shows a larger average eigenvalue compared with the best standard activation function on the Allen-Cahn equation.

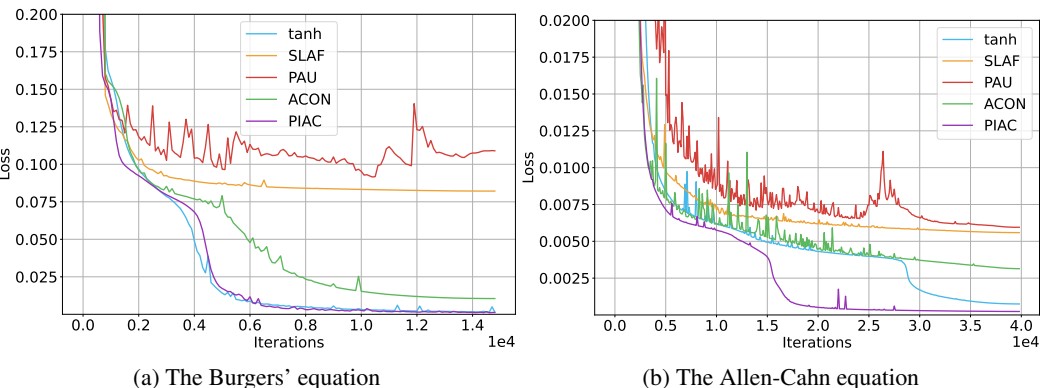

(a) The Burgers' equation        (b) The Allen-Cahn equation

Figure 2: The training loss of different activation functions.

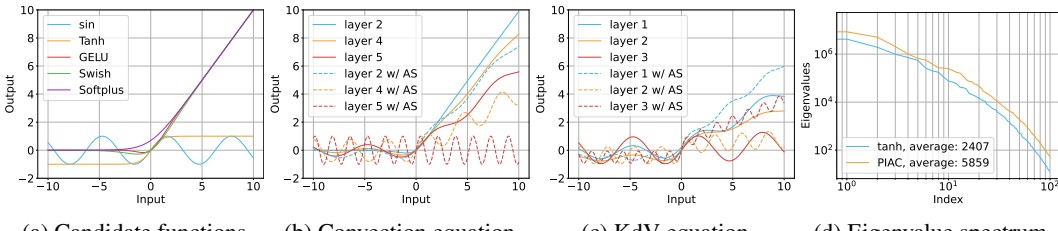

(a) Candidate functions    (b) Convection equation    (c) KdV equation    (d) Eigenvalue spectrum

Figure 3: Visualization of the learned activation functions and the eigenvalue spectrum. (a) Candidate functions of PIAC. (bc) The learned activation functions for the convection equation and the KdV equation. The curves of PIAC with and without the adaptive slope are presented. (d) The eigenvalue spectrum of NTK matrix on the training data of the Allen-Cahn equation.

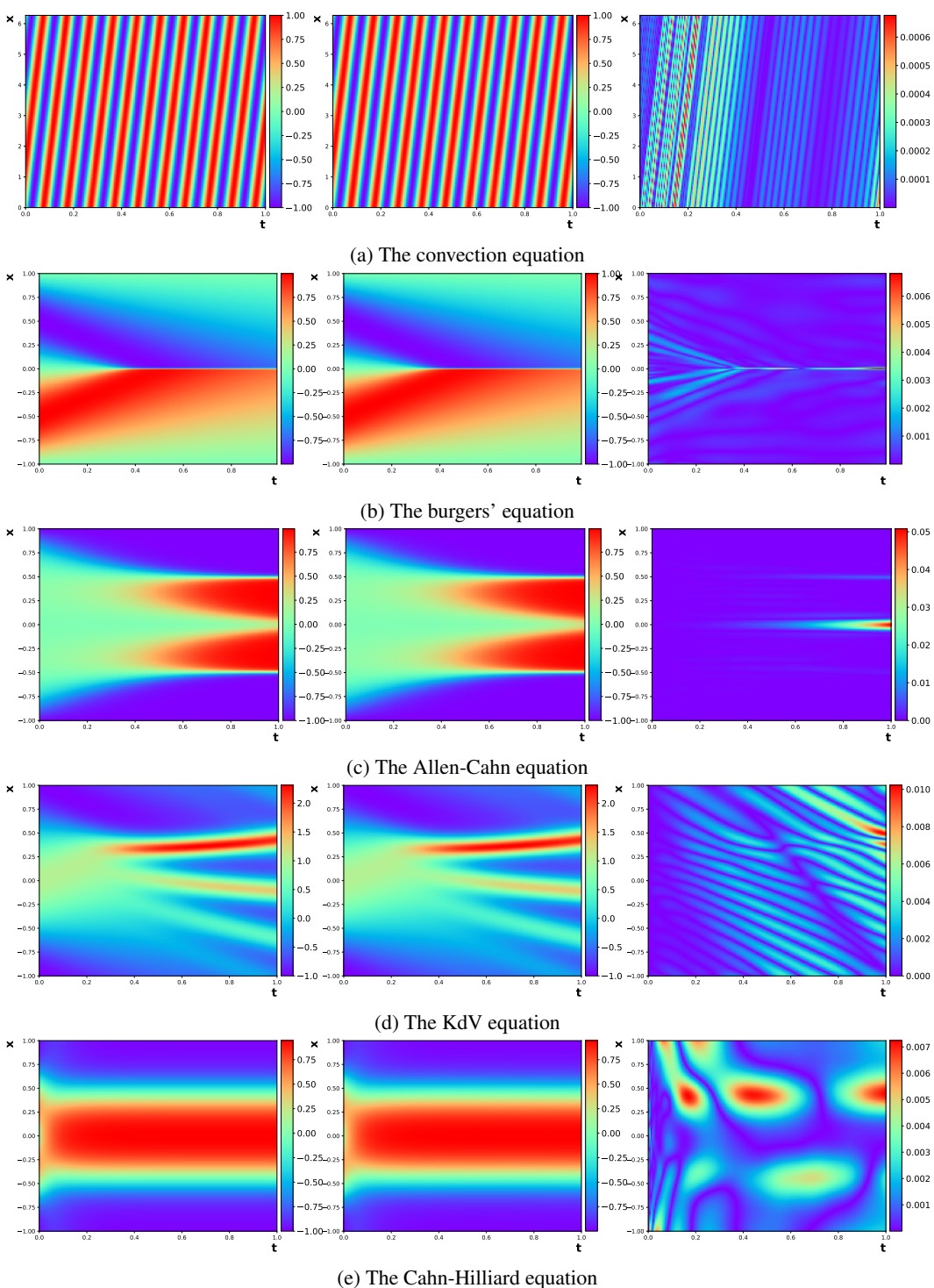

Figure 4: Visualizations of the predictions of PIAC. For each problem, we show the exact solution (left), the prediction (middle) and the absolute error between them (right).

