# OpenReview forum: "Learning Specialized Activation Functions for Physics-informed Neural Networks"
_ICLR.cc/2023/Conference — Submitted to ICLR 2023_

### Official Review · Reviewer_E27E · 2022-10-14

**Confidence:** 5
**Correctness:** 1
**Technical Novelty And Significance:** 2
**Empirical Novelty And Significance:** 1
**Recommendation:** 3

**Clarity, Quality, Novelty And Reproducibility:**

The paper is written well except for a small number of typos (e.g., "over-parameterizaiton"). The experimental part is well detailed but with an insufficient number of comparisons. Novelty is extremely small, as described above.

**Strength And Weaknesses:**

Strength: From the point of view of PINNs, the proposed neural networks have better accuracy and performance than the base ones.

Drawbacks:
- From the methodological point of view, the method is *equivalent* to the method proposed in (Manessi and Rozza, 2019), referenced in the paper, which was also proposed under the name of "adaptive blending unit" in (Sutfeld et al., 2020), which is not referenced here. The authors are renominating them as "physics-informed activation function" (PIAC), but as far as I can see there is no difference that warrants a change of name (apart from slightly modifying the number of base AFs). They mention that "previous methods experiment with limit choice of activation functions", but they use 5 base AFs which is similar to 6 base AFs in (Sutfeld et al., 2020).
- The method is also compared only with fixed or parametric AFs, and not against similar expressive AFs (e.g., APLs).
- The NTK derivation is quite shallow and it does not provide any special insight into the AFs themselves.

**Summary Of The Paper:**

The paper proposes trainable activation functions for physics-informed neural networks (PINNs). The basic idea is to consider a set of base activation functions, and build a trainable one as a convex combination of the base ones. They provide experiments on many problems, showing that PINNs are generally sensitive to the choice of AF, while the trainable one provides good performance across most cases. They also try to motivate the approach from the point of view of the neural tangent kernel (NTK).

**Summary Of The Review:**

The method is a rebranding of a known technique, applied on a different domain. While this is potentially interesting for PINNs, I feel this is not a good practice and it is also an artificial construct to increase novelty.

---

> ### Author Response · Authors · 2022-11-10
> **Response to Reviewer E27E**
>
> Thank you for the valuable comments. We are glad that the reviewer found our approach to be effective from the view of PINNs. Our responses to the comments are as follows.
>
> > **Comment #1**: Novelty is extremely small, as described above.
>
> **Reply #1**:
>
> Thank you for the comment, but we cannot fully agree with the comment. We would like to emphasize that our goal is to introduce the idea of learning combinations of activation functions to PINNs to avoid the inefficient manual selection of activation functions and to alleviate the optimization difficulty of PINNs, rather than proposing a new learnable activation function. While similar ideas have been studied for CNNs in image classification (Dushkoff & Ptucha, 2016; Qian et al., 2018; Manessi & Rozza, 2018;  S ̈utfeld et al., 2020), some technical challenges remain unexplored in the context of PINNs, which have a higher demand for the smoothness and diversity of the candidate functions. Our work overcomes these challenges and evaluates the proposed method through various PDEs. We use the name of "physics-informed activation functions" to emphasize its application to PINNs. We refer to the comment of Reviewer J89g: "The approach is somewhat novel in the PINN literature".
>
> Different from image classification, learning combinations of activation functions in the context of PINNs faces new challenges arising from the peculiar nature of the problem. First, the optimization of PDE-based constraints needs the activation function to provide higher-order derivatives, which causes the failure of widely-used ReLUs and other piecewise linear functions in PINNs. Second, unlike the image classification tasks, different PDE systems could have various characteristics, such as periodicity and rapid decay. This leads to a higher requirement for the diversity of the candidate functions. To overcome these challenges, we propose to build the candidate function set with simple elementary functions to embed the prior knowledge of physics systems, as well as commonly-used activation functions to ensure the diversity.  The effectiveness of our method is demonstrated in a variety of PDE systems. We have made these technical challenges clear in our revision.
>
> Another contribution of our work to the PINN community is to shed light on the relationship between the optimization difficulty of PINN and activation functions, which is also less explored in previous works. Please refer to Section 2.1 for a detailed discussion. Taking the convection equation as an example, Krishnapriyan et al. (2021) finds that vanilla PINNs with tanh (a commonly-used activation function) have difficulty in solving this problem when the convection coefficient is high. They propose a curriculum learning strategy to tackle this optimization difficulty. Without modifying the training strategy, we find the convergence issue of vanilla PINN can be eliminated by selecting a suitable activation function and its performance can be further improved with our method.
>
> We have added the reference to S ̈utfeld et al., 2020 in our revision and removed the inappropriate statement about the number of base AFs in previous works.
>
> > **Comment #2**: The method is also compared only with fixed or parametric AFs, and not against similar expressive AFs (e.g., APLs).
>
> **Reply #2**:
>
> Thank you for the suggestion. We add the comparison to other learnable activation functions (please refer to **Respone to Reviewer qn6w**). However, we do not include APLs (Agostinelli et al., 2014) in the comparison since they are based on piecewise linear functions which cannot provide higher order derivatives and are not suitable for PINNs.
>
> > **Comment #3**: The NTK derivation is quite shallow and it does not provide any special insight into the AFs themselves.
>
> **Reply #3**:
>
> Thank you for pointing out this issue. We acknowledge that it is an important and promising research direction to provide a rigorous theoretical analysis of our method to give special insight into the activation functions. In the current analysis, it is difficult for us to derive the effect of PIAC on NTK theoretically, and we have to analyze it through experimental results. This is motivated by prior works (Wang et al., 2022a;b). For instance, Wang et al. (2022b) relates the training difficulty of PINN to the empirically observed discrepancy between eigenvalues of NTKs corresponding to different loss terms. We would like to emphasize that in this work we focus on the experimental evaluation of the proposed method and the NTK based analysis aims to provide some intuitions to better understand our method. We will explore the rigorous theoretical analysis in future work.
>
> > **Comment #4**: The paper is written well except for a small number of typos (e.g., "over-parameterizaiton").
>
> **Reply #4**:
>
> Thanks for pointing out the typos. We have fixed them in the revision.

---

> > ### Author Response · Authors · 2022-11-10
> > **Response to Reviewer E27E**
> >
> > **References**
> >
> > Forest Agostinelli, Matthew Hoffman, Peter Sadowski, and Pierre Baldi. Learning activation functions to improve deep neural networks. arXiv preprint arXiv:1412.6830, 2014.
> >
> > Michael Dushkoff and Raymond Ptucha. Adaptive activation functions for deep networks. Electronic Imaging, 2016(19):1–5, 2016.
> >
> > Sheng Qian, Hua Liu, Cheng Liu, Si Wu, and Hau San Wong. Adaptive activation functions in convolutional neural networks. Neurocomputing, 272:204–212, 2018.
> >
> > Franco Manessi and Alessandro Rozza. Learning combinations of activation functions. In 2018 24th international conference on pattern recognition (ICPR), pp. 61–66. IEEE, 2018.
> >
> > Leon Ren ́e S ̈utfeld, Flemming Brieger, Holger Finger, Sonja F ̈ullhase, and Gordon Pipa. Adaptive blending units: Trainable activation functions for deep neural networks. In Science and Informa-tion Conference, pp. 37–50. Springer, 2020.
> >
> > Aditi Krishnapriyan, Amir Gholami, Shandian Zhe, Robert Kirby, and Michael W Mahoney. Characterizing possible failure modes in physics-informed neural networks. Advances in Neural Information Processing Systems, 34, 2021.
> >
> > Sifan Wang, Shyam Sankaran, and Paris Perdikaris. Respecting causality is all you need for training physics-informed neural networks. arXiv preprint arXiv:2203.07404, 2022a.
> >
> > Sifan Wang, Xinling Yu, and Paris Perdikaris. When and why pinns fail to train: A neural tangent kernel perspective. Journal of Computational Physics, 449:110768, 2022b.

---

> > > ### Comment · Reviewer_E27E · 2022-11-21
> > > **Response to the authors**
> > >
> > > Thanks for the comprehensive answer. I do agree with the authors that the problem of selecting proper activation functions is crucial in PINNs. I also agree that the field of trainable AFs is underexplored in this regard.
> > >
> > > However, I do not agree that changing the application field warrants a change in name of a method. The  physics-informed activation function (PIAC, Eq. (10)), is **identical** to, among others, the soft-normalized version of the adaptive blending unit (ABU) [1], see the second sentence of Section 3.1 in [1] and the $\text{ABU}_{\text{soft}}$ variant in Section 4.1.
> > >
> > > The authors state that they "*propose to build the candidate function set with simple elementary functions to embed the prior knowledge of physics systems*". However, in the experiments we read that "*We set the candidate function set F as {sin,tanh, GELU, Swish, Softplus}*". In ABU [1] we read "*The activation functions we used as a baseline throughout this work are the hyperbolic tangent (tanh), ReLU, ELU, SELU, the identity and Swish*". The only difference is adding a sin function to the baseline set.
> > >
> > > Hence, the contribution of this work is applying ABUs to PINNs; the merits of this can be discussed, but choosing to modify a name to try and increase the novelty is a very poor practice and because of this I prefer not to modify my own score.
> > >
> > > [1] https://arxiv.org/pdf/1806.10064.pdf

---

> > > > ### Author Response · Authors · 2022-11-22
> > > > **Additional Response to Reviewer E27E**
> > > >
> > > > Thanks for your feedback. As you mentioned, the goal of our work is to study adaptive activation functions in the context of PINNs. Indeed, many adaptive activation functions exist in the general deep learning literature, some of which have similar forms as our method (Dushkoff & Ptucha, 2016; Qian et al., 2018; Manessi & Rozza, 2018;  S ̈utfeld et al., 2020). However, the subtle difference between those works and our method is exactly where our contribution lies.
> > > >
> > > > Taking the example you pointed out, the candidate function set of ABU (S ̈utfeld et al., 2020) and that of our method appears to be very similar. Superficially, we only added the sin function to the set, while removing the ReLU function. But in fact, these subtle differences highlight our special treatments tailored for PINNs optimization.
> > > >
> > > > First, we use the sin function to embed the periodicity, a common characteristic of physic systems but rarely been considered in other contexts (CNN, RNN, Transformers, etc),  into neural networks. Note that the idea behind this is quite general: we can incorporate elementary functions with different properties according to our prior knowledge of a physics system at hand. For instance, we also use the exponential function to help the learning of rapid decay when solving 1D Poisson's equations (see Appendix C). None of the existing works (Dushkoff & Ptucha, 2016; Qian et al., 2018; Manessi & Rozza, 2018;  S ̈utfeld et al., 2020) have considered adding these functions to their candidate set, because there is no such prior knowledge to leverage in general domains like image classification or natural language processing. In contrast, we reveal that adaptive activation functions can be more powerful and interpretable in handling PDE systems.
> > > >
> > > > Second, we remove those activation functions which cannot provide continuous and non-zero higher-order derivatives, such as ReLU, ELU and identity. These functions may hinder the application of adaptive activation functions to PINNs whose objective function includes PDE-based constraints.
> > > >
> > > > In summary, our main contribution is to remove obstacles to apply adaptive activation functions to PINNs. It provides a novel solution to alleviate the optimization difficulty of PINNs from the new perspective of network (micro) architecture design. We hope that our work could bring some insights and practical tools to the PINNs community, as the other three reviewers acknowledged.
> > > >
> > > > We do agree that the name of "physics-informed activation functions" may distract the readers from the main contribution of our work.  Since we cannot update our draft now, we promise to rephrase the Introduction section to highlight our contribution and give more credits to previous works on adaptive activation functions.
> > > >
> > > > We hope our response could resolve your concerns. Thanks for your time.
> > > >
> > > > **Reference**
> > > >
> > > > Michael Dushkoff and Raymond Ptucha. Adaptive activation functions for deep networks. Electronic Imaging, 2016(19):1–5, 2016.
> > > >
> > > > Sheng Qian, Hua Liu, Cheng Liu, Si Wu, and Hau San Wong. Adaptive activation functions in convolutional neural networks. Neurocomputing, 272:204–212, 2018.
> > > >
> > > > Franco Manessi and Alessandro Rozza. Learning combinations of activation functions. In 2018 24th international conference on pattern recognition (ICPR), pp. 61–66. IEEE, 2018.
> > > >
> > > > Leon Ren ́e S ̈utfeld, Flemming Brieger, Holger Finger, Sonja F ̈ullhase, and Gordon Pipa. Adaptive blending units: Trainable activation functions for deep neural networks. In Science and Informa-tion Conference, pp. 37–50. Springer, 2020.

---

> > > > > ### Comment · Reviewer_E27E · 2022-11-22
> > > > > **Additional response**
> > > > >
> > > > > *some of which have similar forms as our method*: I disagree on this sentence. They are not similar, they are **identical**, as you are also mentioning. The difference is in the candidate set, which is not reason enough to change the name of the method. I agree that the analysis is tailored for PINNs and I also agree it is correct. Again, I disagree that ABUs and PIACs are different objects requiring different names.
> > > > >
> > > > > *the name of "physics-informed activation functions" may distract the readers*: it is not a question of being distracting, it is a question of being disrespectful to other researchers by copying their methods, modifying a non-essential part, and claiming it to be novel to artificially inflate the novelty.
> > > > >
> > > > > Once again, I am not discussing the specific merits of applying trainable AFs to the field of PINNs. However, I believe that name changing is not up to the ICLR Code of Ethics.

---

> > > > > > ### Author Response · Authors · 2022-11-23
> > > > > > **Additional Response to Reviewer E27E**
> > > > > >
> > > > > > Thanks for your response. We would like to clarify that the name of "physics-informed activation functions" was not intended to differentiate our method from ABU (S ̈utfeld et al., 2020) or any other related methods (Dushkoff & Ptucha, 2016; Qian et al., 2018; Manessi & Rozza) to exaggerate the novelty. In fact, we were not aware of the ABU paper (S ̈utfeld et al., 2020) before the reviewer mentioned it. In the original version of our submission, we list three works (Dushkoff & Ptucha, 2016; Qian et al., 2018; Manessi & Rozza, 2018) to point out that the idea of learning combinations of activation functions has been explored for CNN in image classification and our main contribution is to introduce this idea into PINNs. Since these three works have no unified name, we used PIAC for convenience and easy reference.
> > > > > >
> > > > > > We agree that giving a new name to an exisiting approach is disrespectful for the original researchers. As we promised in our last response, we will revise our paper (possibly replacing the name PIAC with PINN-ABU) to give more credits to previous works on adaptive activation functions. But at this stage, we are not able to update the draft in the submission system.
> > > > > >
> > > > > > We thank the reviewer for mentioning the work of ABU so that we can make the literature review more complete. Again, we would like to note that this existing work should not degrade our main contribution, as we have discussed above.
> > > > > >
> > > > > > **Reference**
> > > > > >
> > > > > > Michael Dushkoff and Raymond Ptucha. Adaptive activation functions for deep networks. Electronic Imaging, 2016(19):1–5, 2016.
> > > > > >
> > > > > > Sheng Qian, Hua Liu, Cheng Liu, Si Wu, and Hau San Wong. Adaptive activation functions in convolutional neural networks. Neurocomputing, 272:204–212, 2018.
> > > > > >
> > > > > > Franco Manessi and Alessandro Rozza. Learning combinations of activation functions. In 2018 24th international conference on pattern recognition (ICPR), pp. 61–66. IEEE, 2018.
> > > > > >
> > > > > > Leon Ren ́e S ̈utfeld, Flemming Brieger, Holger Finger, Sonja F ̈ullhase, and Gordon Pipa. Adaptive blending units: Trainable activation functions for deep neural networks. In Science and Informa-tion Conference, pp. 37–50. Springer, 2020.

---

> ### Author Response · Authors · 2022-11-29
> **A kind reminder to Reviewer E27E**
>
> Dear Reviewer E27E,
>
> We would like to thank you again for your time in reviewing our work. As the deadline for discussion is approaching, we really hope to have a further discussion with you to see if our response solves the concerns. Please do not hesitate to contact us if you have any remaining concerns or questions.
>
> Best wishes,
>
> Authors

---

> > ### Comment · Reviewer_E27E · 2022-11-29
> > **Answer to the reminder**
> >
> > As I wrote, my score does not reflect the merits of applying trainable AFs to PINNs, which may be valuable (in particular, I have no concerns on the experimental evaluation). My score reflects the naming decision, which in my opinion is poor practice and it is also against any reasonable code of ethics. Since the paper is built around the idea of introducing a novel AF (which is objectively not true), not on properly modifying known trainable AFs, no amount of revision can modify it. Hence, I prefer to keep my score and simply leave this as a matter of discussion for the area chairs. In particular, since related works were cited in the original version of the paper, I assume the naming decision was intentional.

---

### Official Review · Reviewer_J89g · 2022-10-22

**Confidence:** 3
**Correctness:** 3
**Technical Novelty And Significance:** 3
**Empirical Novelty And Significance:** 3
**Recommendation:** 6

**Clarity, Quality, Novelty And Reproducibility:**

The paper is well written and clear.
One thing that was not clear to me is: Do all activation functions in the network share the same parameters? or are they all learned independently?
The approach is somewhat novel in the PINN literature.
The work appears to be reasonably reproducible. I have no concerns there.

The other reviewers have raised the criticism that the proposed method is very similar to existing work. I was not aware of that work when I wrote my review. I find that the existence of the prior work significantly reduces my view of the novelty of the authors' contribution. I have therefore reduced my score. I think the authors could improve their contribution by formally proving their NTK claims or perhaps by expanding the number of base activation functions to very large numbers (e.g. using a basis).

**Strength And Weaknesses:**

The method proposed by the authors addresses a significant issue in PINNs. The approach is demonstrated on 5 different PDEs. The method is compared to several baselines as well as some ablations.

The NTK based analysis is quite hand-wavy. There is no treatment of the approximation error of the two-phase analysis and the notion that the NTK is learned to suite the underlying PDE is approached empirically with a single example.

**Summary Of The Paper:**

The authors present a method for parameterizing activation functions to improve convergence in physics-informed-neural-networks. They demonstrate through empirical experimentation that the proposed method converges to better solutions than non-learned activation functions or adaptive slope methods. Finally, the authors provide an analysis based on Neural Tangent Kernel that sheds light on why the proposed method is effective.


**Summary Of The Review:**

Overall this paper is a solid demonstration of an approach to address the training difficulty empirically observed in PINNs.

---

> ### Author Response · Authors · 2022-11-10
> **Response to Reviewer J89g**
>
> Thank you for the valuable comments. We are encouraged that the reviewer found our approach to be novel in PINN literature and address a significant issue in PINNs. Our responses to the comments are as follows.
>
> > **Comment #1**: Do all activation functions in the network share the same parameters? or are they all learned independently?
>
> **Reply #1**:
>
> As mentioned in Section 4.1, we apply the proposed method in a layer-wise manner, which means the parameters of activation functions in different layers are learned independently while those in the same layer are shared.
>
> > **Comment #2**: The NTK based analysis is quite hand-wavy. There is no treatment of the approximation error of the two-phase analysis and the notion that the NTK is learned to suite the underlying PDE is approached empirically with a single example.
>
> **Reply #2**:
>
> Thank you for pointing out this issue. We acknowledge that it is an important and promising research direction to provide a rigorous theoretical analysis to deal with the approximation error of the two-phase assumption and to support the notion that the NTK is learned to suit the underlying PDE. In the current analysis, it is difficult for us to derive the effect of PIAC on NTK theoretically, and we have to analyze it through experimental results. This is motivated by prior works (Wang et al., 2022a;b). For instance, Wang et al. (2022b) relates the training difficulty of PINN to the empirically observed discrepancy between eigenvalues of NTKs corresponding to different loss terms. We would like to emphasize that in this work we focus on the experimental evaluation of the proposed method and the NTK based analysis aims to provide some intuitions to better understand our method. We will explore the rigorous theoretical analysis in future work.
>
> **References**
>
> Sifan Wang, Shyam Sankaran, and Paris Perdikaris. Respecting causality is all you need for training physics-informed neural networks. arXiv preprint arXiv:2203.07404, 2022a.
>
> Sifan Wang, Xinling Yu, and Paris Perdikaris. When and why pinns fail to train: A neural tangent kernel perspective. Journal of Computational Physics, 449:110768, 2022b.

---

### Official Review · Reviewer_qn6w · 2022-10-25

**Confidence:** 5
**Correctness:** 4
**Technical Novelty And Significance:** 3
**Empirical Novelty And Significance:** 2
**Recommendation:** 8

**Clarity, Quality, Novelty And Reproducibility:**

The paper is well written and technically sound. The gating function and components are presented clearly. The activation function can be reproduced from the information in the paper.

**Strength And Weaknesses:**

The authors present a technically sound approach for PINNs. Evaluated on datasets that clearly present the difficulties in this domain.

However, the main weakness of this paper is in the lack of comparison to other learnable activation functions. The authors present these family of activation functions in the section "Adaptive activation functions", yet they are not present in the evaluation.
Some of them could be relevant as they do provide higher oder derivatives. Maybe there is an issue with those activation functions and that is a good reason not to include them, but this is not clear from the paper.



**Summary Of The Paper:**

In this paper, the authors present a formulation to create a trainable activation function based on a convex combination of other activations.
This new mixture approach is intended to work on physics-informed neural networks (PINNd) where higher order derivatives play a role, thus causing problems for some of the piecewise linear based common activations.

The authors the provide an empirical evaluation as well as a link to the neural tangent kernel.

**Summary Of The Review:**

In this paper, the authors present an activation function that is amenable to PINNs. The authors present the problem clearly and offer a solution for this particular context based on previous work. They however do not compare to other adaptable activation functions that might still behave well in this domain.
The empirical evaluation would be much stronger, if you were to compare to that class of activations. In particular, it would be great to show the benefits of your activation function, or explain why the others are not suitable and demonstrate that empirically.

---

> ### Author Response · Authors · 2022-11-10
> **Response to Reviewer qn6w**
>
> Thank you for the valuable comments. We are glad that the reviewer found our motivation to be clear and our method to be technically sound. The main concern is the lack of comparison to other learnable activation functions. We provide this comparison as follows.
>
> We give a brief review of learnable activation functions in Section 2.2, some of which are not suitable for PINNs because their second-order derivatives are zero everywhere, such as APL (Agostinelli et al., 2014), RePLU (Li et al., 2016) and PWLU (Zhou et al., 2021). We compare our method to other learnable activation functions which could provide higher-order derivatives, including SLAF (Goyal et al., 2019), PAU (Molina et al., 2019) and ACON (Ma et al., 2021). Their formulations are shown as follows.
>
> 1. The SLAF proposes to learn activation functions with Taylor polynomial bases, which is presented as
> $$f(x)=\sum_{i=0}^ma_ix^i,$$
> where m is set to 5 in our experiments. The parameters are initialized from a normal distribution.
> 2. The PAU is presented as the rational functions of the form
> $$f(x)=\frac{\sum_{i=0}^ma_ix^i}{1+|\sum_{j=1}^nb_jx^j|},$$
> where m is set to 5 and n is set to 4 following the default setting in Molina et al., 2019. The absolute value is to avoid a zero-valued denominator. The parameters are initialized from a normal distribution.
> 3. The ACON is a smooth approximator to the general Maxout family activation functions, which is presented as
> $$f(x)=(p_1-p_2)x*\mathrm{sigmoid}(\beta(p_1-p_2)x)+p_2x,$$
> Where $p_1$, $p_2$ and $\beta$ are learnable parameters and initialized from a normal distribution.
>
> The results of different learnable activation functions are shown in the following table. We use the same experimental configurations described in Appendix D. The $L_2$ relative error (%) is reported. SLAF achieves performance comparable to the best standard activation function on the first-order convection equation, but does not produce accurate predictions for other higher-order PDEs. Although SLAF shares a similar formulation with PIAC, its polynomial bases can cause vanishing or exploding gradients due to the high order powers in its derivatives. This problem hinders the optimization of higher-order PDE-based constraints. PAU suffers from instability of training (see Appendix Figure 2) and performs poorly on all PDEs. We attribute the training instability of PAU to its discontinuous derivatives, which arise from the absolute value in the denominator of its formulation. ACON achieves stable performance but does not surpass the best standard activation function in each problem due to the limited flexibility of its formulation. The proposed method outperforms these learnable activation functions thanks to the diverse and smooth candidate functions, which can be used to incorporate the prior knowledge of different physics systems.
>
> We have added the comparison and discussion in our revision.
>
> |Method|Convection|Burgers'|Allen-Cahn|KdV|Cahn-Hilliard|Average error|
> | :----: | :----: | :----: | :----: | :----: | :----: | :----: |
> |sin|$0.36\pm0.15$|$5.18\pm3.73$|$3.57\pm0.64$|$0.63\pm0.17$|$1.72\pm0.61$|2.29|
> |tanh|$6.83\pm4.79$|$0.26\pm0.13$|$1.34\pm0.54$|$1.32\pm1.12$|$4.02\pm4.56$|2.75|
> |SLAF|$0.36\pm0.18$|$43.93\pm0.66$|$33.93\pm9.97$|$25.23\pm1.28$|$52.57\pm27.93$|31.20|
> |PAU|$45.78\pm35.47$|$48.31\pm8.21$|$43.83\pm15.18$|$68.11\pm13.49$|$115.59\pm3.79$|64.32|
> |ACON|$3.55\pm1.66$|$1.18\pm1.55$|$3.88\pm1.82$|$1.52\pm0.35$|$2.46\pm1.96$|2.52|
> |PIAC|$\mathbf{0.06\pm0.03}$|$\mathbf{0.21\pm0.11}$|$\mathbf{0.76\pm0.26}$|$\mathbf{0.34\pm0.05}$|$\mathbf{0.50\pm0.15}$|$\mathbf{0.37}$|
>
> **References**
>
> Forest Agostinelli, Matthew Hoffman, Peter Sadowski, and Pierre Baldi. Learning activation functions to improve deep neural networks. arXiv preprint arXiv:1412.6830, 2014.
>
> Jun Li, Tong Zhang, Wei Luo, Jian Yang, Xiao-Tong Yuan, and Jian Zhang. Sparseness analysis in the pretraining of deep neural networks. IEEE transactions on neural networks and learning systems, 28(6):1425–1438, 2016.
>
> Mohit Goyal, Rajan Goyal, and Brejesh Lall. Learning activation functions: A new paradigm of understanding neural networks. arxiv 2019. arXiv preprint arXiv:1906.09529, 2019.
>
> Alejandro Molina, Patrick Schramowski, and Kristian Kersting. Pade activation units: End-to-end ´ learning of flexible activation functions in deep networks. arXiv preprint arXiv:1907.06732, 2019.
>
> Yucong Zhou, Zezhou Zhu, and Zhao Zhong. Learning specialized activation functions with the piecewise linear unit. In Proceedings of the IEEE/CVF International Conference on Computer Vision, pp. 12095–12104, 2021.
>
> Ningning Ma, Xiangyu Zhang, Ming Liu, and Jian Sun. Activate or not: Learning customized activation. In Proceedings of the IEEE/CVF Conference on Computer Vision and Pattern Recognition, pp. 8032–8042, 2021.

---

> > ### Comment · Reviewer_qn6w · 2022-11-16
> > **Interesting results**
> >
> > Given these new interesting results, I feel confident in raising the score of my review.

---

> > > ### Author Response · Authors · 2022-11-22
> > > **Thank you for your feedback**
> > >
> > > Thank you for your time and support! We sincerely appreciate your valuable comments and suggestions, which helped us improve the quality of the draft.

---

### Official Review · Reviewer_5qLY · 2022-10-25

**Confidence:** 4
**Correctness:** 4
**Technical Novelty And Significance:** 2
**Empirical Novelty And Significance:** 2
**Recommendation:** 6

**Clarity, Quality, Novelty And Reproducibility:**

The paper is well written and organized. However, the technical innovation is limited.

**Strength And Weaknesses:**

* Strength
1. It is interesting to automatically choose the activation functions for PINN
2. The proposed method is simple yet effective to solve PDE systems
3. Conduct extensive experiments on various PDEs to verify the effectiveness of the proposed approach

* Limitations
1. Technical innovation is somewhat limited. The proposed gate function to learn the coefficients of activation functions is the same as that in the literature [Qian et al. 2018]
2. Compare to more baselines. I am curious if you could compare the prior works that learn combinations of activation functions?
3. A few grammatical errors. For example, "by minimize the following objective function" in the bottom of page 2; and "we consider a extreme case of insufficient collocation points", a-> an

References:\
[Qian et al. 2018] Adaptive activation functions in convolutional neural networks, 2018.

**Summary Of The Paper:**

The goal of this work is to automatically learn the activation functions via the combinations of different candidate activation functions for PINN. Specifically, it adopted gate function with a learnable parameter to identify the coefficients for different candidate activation functions. Extensive experiments are carried out to evaluate the proposed physics-informed activation functions on a variety of partial differential equations (PDEs). The results demonstrate the efficacy of the simple approach.

**Summary Of The Review:**

It is interesting to automatically learn the activation functions for PINNs. The proposed method is simple yet effective for PDEs systems. However, this work adopted the same gate function method from prior work and apply it to PINNs. Hence, the technical innovation is somewhat limited.

---

> ### Author Response · Authors · 2022-11-10
> **Response to Reviewer 5qLY**
>
> Thank you for the valuable comments. We are encouraged that the reviewer found learning activation functions for PINNs to be an interesting idea and our method to be effective. Our responses to the comments are as follows.
> > **Comment #1**: Technical innovation is somewhat limited. The proposed gate function to learn the coefficients of activation functions is the same as that in the literature [Qian et al. 2018].
>
> **Reply #1**:
>
> Thank you for the comment. We would like to emphasize that our goal is to introduce the idea of learning combinations of activation functions to PINNs to avoid the inefficient manual selection of activation functions and to alleviate the optimization difficulty of PINNs, rather than proposing a new learnable activation function. While similar ideas have been studied for CNNs in image classification (Dushkoff & Ptucha, 2016; Qian et al., 2018; Manessi & Rozza, 2018;  S ̈utfeld et al., 2020), some technical challenges remain unexplored in the context of PINNs, which have a higher demand for the smoothness and diversity of the candidate functions. Our work overcomes these challenges and evaluates the proposed method through various PDEs. We refer to the comment of Reviewer J89g: "The approach is somewhat novel in the PINN literature".
>
> Different from image classification, learning combinations of activation functions in the context of PINNs faces new challenges arising from the peculiar nature of the problem. First, the optimization of PDE-based constraints needs the activation function to provide higher-order derivatives, which causes the failure of widely-used ReLUs and other piecewise linear functions in PINNs. Second, unlike the image classification tasks, different PDE systems could have various characteristics, such as periodicity and rapid decay. This leads to a higher requirement for the diversity of the candidate functions. To overcome these challenges, we propose to build the candidate function set with simple elementary functions to embed the prior knowledge of physics systems, as well as commonly-used activation functions to ensure the diversity.  The effectiveness of our method is demonstrated in a variety of PDE systems. We have made these technical challenges clear in our revision.
>
> Another contribution of our work to the PINN community is to shed light on the relationship between the optimization difficulty of PINN and activation functions, which is also less explored in previous works. Please refer to Section 2.1 for a detailed discussion. Taking the convection equation as an example, Krishnapriyan et al. (2021) finds that vanilla PINNs with tanh (a commonly-used activation function) have difficulty in solving this problem when the convection coefficient is high. They propose a curriculum learning strategy to tackle this optimization difficulty. Without modifying the training strategy, we find the convergence issue of vanilla PINN can be eliminated by selecting a suitable activation function and its performance can be further improved with our method.
>
> > **Comment #2**: Compare to more baselines. I am curious if you could compare the prior works that learn combinations of activation functions?
>
> **Reply #2**:
>
> Thank you for this suggestion. The main differences between prior works that learn combinations of activation functions (Dushkoff & Ptucha, 2016; Qian et al., 2018; Manessi & Rozza, 2018; Goyal et al., 2019;  S ̈utfeld et al., 2020) lie in gate functions and candidate functions. Most works (Dushkoff & Ptucha, 2016; Qian et al., 2018; Manessi & Rozza, 2018;  S ̈utfeld et al., 2020) select candidate functions from commonly-used activation functions, while SLAF learns a weighted sum of Taylor polynomial bases. For the former, we refer to the Table 4 which compares different choices of gate functions and candidate function set; for the latter, we refer to **Response to Reviewer qn6w** where we compare our methods to other learnable activation functions.
>
> > **Comment #3**: A few grammatical errors. For example, "by minimize the following objective function" in the bottom of page 2; and "we consider a extreme case of insufficient collocation points", a-> an.
>
> **Reply #3**:
>
> Thank you for pointing out these grammatical errors. We have fixed them in the revision.

---

> > ### Author Response · Authors · 2022-11-10
> > **Response to Reviewer 5qLY**
> >
> > **References**
> >
> > Michael Dushkoff and Raymond Ptucha. Adaptive activation functions for deep networks. Electronic Imaging, 2016(19):1–5, 2016.
> >
> > Sheng Qian, Hua Liu, Cheng Liu, Si Wu, and Hau San Wong. Adaptive activation functions in convolutional neural networks. Neurocomputing, 272:204–212, 2018.
> >
> > Franco Manessi and Alessandro Rozza. Learning combinations of activation functions. In 2018 24th international conference on pattern recognition (ICPR), pp. 61–66. IEEE, 2018.
> >
> > Mohit Goyal, Rajan Goyal, and Brejesh Lall. Learning activation functions: A new paradigm of understanding neural networks. arxiv 2019. arXiv preprint arXiv:1906.09529, 2019.
> >
> > Leon Ren ́e S ̈utfeld, Flemming Brieger, Holger Finger, Sonja F ̈ullhase, and Gordon Pipa. Adaptive blending units: Trainable activation functions for deep neural networks. In Science and Informa-tion Conference, pp. 37–50. Springer, 2020.
> >
> > Aditi Krishnapriyan, Amir Gholami, Shandian Zhe, Robert Kirby, and Michael W Mahoney. Characterizing possible failure modes in physics-informed neural networks. Advances in Neural Information Processing Systems, 34, 2021.

---

> > > ### Comment · Reviewer_5qLY · 2022-11-20
> > > **Thank you for your response**
> > >
> > > Thank you very much for your response! I have increased my score. I think the application is very interesting, but the technical contribution is not so novel. Thanks a  lot!

---

> > > > ### Author Response · Authors · 2022-11-22
> > > > **Thank you for your feedback**
> > > >
> > > > Thank you for your time and efforts! We sincerely appreciate your valuable comments and suggestions, which helped us improve the quality of the draft.

---

### Decision · Program_Chairs · 2023-01-20

**Decision:**

Reject

**Justification For Why Not Higher Score:**

The idea of learning activation functions is not new and has been proposed in multiple papers.

**Justification For Why Not Lower Score:**

The experimental results are good and the idea can be useful for training PINNs.

**Metareview: Summary, Strengths And Weaknesses:**

The paper proposes to learn an adaptive activation function (as a weighted sum of candidate functions) inside physics-informed neural networks (PINNs). The idea is simple and it is shown to improve the accuracy of the solution approximated by PINNs on multiple differential equations. This is the major strength of the paper acknowledged by all the reviewers.

During the discussion, the following points were raised by the reviewers:

- The idea of using learnable activation functions is not new and it has been explored in other domains (e.g. image classification). The authors admit that but they argue that the task of solving PDEs often benefits from using types of learnable functions which are different from the ones used in previous works (for example, using sin and exp in the trainable mixture may improve the performance). The authors argue that the idea of learnable activation functions has not been explored in the PINN community.

- Lack of comparison to other learnable activation functions. The authors presented experiments showing that the proposed adaptive activation function works well compared to previously proposed alternatives.

During the virtual meeting, the reviewers reached a general agreement that the results are definitely of interest to the PINN community but unfortunately the novelty is not strong enough for acceptance to ICLR. The reviewers encourage the authors to submit the paper to a venue which is more specialized on physics applications and PINNs (e.g. Journal of Computational Physics). The authors are also encouraged to improve writing by giving more credit to previous works on learnable activation functions and perhaps dampening the claim on the novelty of the proposed solution.

**Summary Of Ac-Reviewer Meeting:**

During the virtual meeting, we (the reviewers and AC) reached a general agreement about the recommendations to the PCs and the authors. We appreciate the simplicity and effectiveness of the proposed idea and the good results obtained in the experimental part. And we think that the results are definitely of interest to the PINN community. However, we believe that the novelty is not strong enough for acceptance to ICLR because very similar ideas have existed in the ML literature: 1) multiple papers propose similar ways of learning the activation functions, 2) periodic activation functions have been proposed in the context of PINNs as well (see, e.g., https://arxiv.org/abs/2006.09661). We think that the paper's contribution (though important) might not be interesting enough for a general ML audience. We encourage the authors to submit the paper to a venue which is more specialized on physics applications and PINNs (e.g. Journal of Computational Physics). We also encourage the authors to improve writing by giving more credit to previous works on learnable activation functions and perhaps dampening the claim on the novelty of the proposed solution.